# Carriers of *ADAMTS13* Rare Variants Are at High Risk of Life-Threatening COVID-19

**DOI:** 10.3390/v14061185

**Published:** 2022-05-29

**Authors:** Kristina Zguro, Margherita Baldassarri, Francesca Fava, Giada Beligni, Sergio Daga, Roberto Leoncini, Lucrezia Galasso, Michele Cirianni, Stefano Rusconi, Matteo Siano, Daniela Francisci, Elisabetta Schiaroli, Sauro Luchi, Giovanna Morelli, Enrico Martinelli, Massimo Girardis, Stefano Busani, Saverio Giuseppe Parisi, Sandro Panese, Carmelo Piscopo, Mario Capasso, Danilo Tacconi, Chiara Spertilli Raffaelli, Annarita Giliberti, Giulia Gori, Peter D. Katsikis, Maria Lorubbio, Paola Calzoni, Agostino Ognibene, Monica Bocchia, Monica Tozzi, Alessandro Bucalossi, Giuseppe Marotta, Simone Furini, Alessandra Renieri, Chiara Fallerini

**Affiliations:** 1Med Biotech Hub and Competence Center, Department of Medical Biotechnologies, University of Siena, 53100 Siena, Italy; kristina.zguro@dbm.unisi.it (K.Z.); margherita.baldassarri@dbm.unisi.it (M.B.); francesca.fava@dbm.unisi.it (F.F.); giada.beligni@dbm.unisi.it (G.B.); sergio.daga@dbm.unisi.it (S.D.); simone.furini@unisi.it (S.F.); fallerini2@unisi.it (C.F.); 2Medical Genetics, University of Siena, 53100 Siena, Italy; 3Genetica Medica, Azienda Ospedaliero-Universitaria Senese, 53100 Siena, Italy; 4Laboratorio Patologia Clinica, Azienda Ospedaliero-Universitaria Senese, 53100 Siena, Italy; rob.leoncini@ao-siena.toscana.it (R.L.); lucrezia.galasso@dbm.unisi.it (L.G.); michele.cirianni@dbm.unisi.it (M.C.); p.calzoni@ao-siena.toscana.it (P.C.); 5Infectious Diseases Unit, ASST Ovest Milanese, 20025 Legnano, Italy; stefano.rusconi@unimi.it; 6Department of Biomedical and Clinical Sciences Luigi Sacco, University of Milan, 20157 Milan, Italy; matteosiano2@gmail.com; 7Infectious Diseases Clinic, “Santa Maria della Misericordia” Hospital, University of Perugia, 06124 Perugia, Italy; daniela.francisci@unipg.it (D.F.); elisabetta.schiaroli@unipg.it (E.S.); 8Infectious Disease Unit, Hospital of Lucca, 55100 Lucca, Italy; sauro.luchi@uslnordovest.toscana.it (S.L.); giovanna.morelli@uslnordovest.toscana.it (G.M.); 9Department of Respiratory Diseases, Azienda Ospedaliera di Cremona, 26100 Cremona, Italy; enrico.martinelli@asst-cremona.it; 10Department of Anesthesia and Intensive Care, University of Modena and Reggio Emilia, 41124 Modena, Italy; girardis@unimore.it (M.G.); stefano.busani@unimore.it (S.B.); 11Department of Molecular Medicine, University of Padova, 35121 Padova, Italy; saverio.parisi@unipd.it; 12Clinical Infectious Diseases, Mestre Hospital, 30171 Venezia, Italy; sandro.panese@aulss3.veneto.it; 13Medical Genetics and Laboratory Genetics Unit, “Antonio Cardarelli” hospital, 80131 Naples, Italy; carmelo.piscopo@aocardarelli.it; 14Department of Molecular Medicine and Medical Biotechnology, University of Naples Federico II, 80138 Naples, Italy; mario.capasso@unina.it; 15CEINGE Biotecnologie Avanzate, 80145 Naples, Italy; 16Department of Specialized and Internal Medicine, Infectious Diseases Unit, San Donato Hospital Arezzo, 52100 Arezzo, Italy; danilo.tacconi@uslsudest.toscana.it (D.T.); chiara.spertilliraffaelli@uslsudest.toscana.it (C.S.R.); 17Medical Genetics Unit, Meyer Children’s University Hospital, 50134 Florence, Italy; annarita.giliberti@meyer.it (A.G.); giulia.gori@meyer.it (G.G.); 18Department of Immunology, Erasmus Medical Center, 3015 GD Rotterdam, The Netherlands; p.katsikis@erasmusmc.nl; 19UOC Laboratorio Analisi Chimico Cliniche, 52100 Arezzo, Italy; maria.lorubbio@uslsudest.toscana.it (M.L.); agostino.ognibene@uslsudest.toscana.it (A.O.); 20Hematology Unit, Department of Medical Science, Surgery and Neuroscience, University of Siena, 53100 Siena, Italy; monica.bocchia@unisi.it; 21Stem Cell Transplant and Cellular Therapy Unit, University Hospital of Siena, 53100 Siena, Italy; m.tozzi@ao-siena.tocana.it (M.T.); alessandro.bucalossi@ao-siena.toscana.it (A.B.); giuseppe.marotta@unisi.it (G.M.)

**Keywords:** COVID-19, *ADAMTS13*, thromboembolism, add-on therapy

## Abstract

Thrombosis of small and large vessels is reported as a key player in COVID-19 severity. However, host genetic determinants of this susceptibility are still unclear. Congenital Thrombotic Thrombocytopenic Purpura is a severe autosomal recessive disorder characterized by uncleaved ultra-large vWF and thrombotic microangiopathy, frequently triggered by infections. Carriers are reported to be asymptomatic. Exome analysis of about 3000 SARS-CoV-2 infected subjects of different severities, belonging to the GEN-COVID cohort, revealed the specific role of vWF cleaving enzyme *ADAMTS13* (A disintegrin-like and metalloprotease with thrombospondin type 1 motif, 13). We report here that ultra-rare variants in a heterozygous state lead to a rare form of COVID-19 characterized by hyper-inflammation signs, which segregates in families as an autosomal dominant disorder conditioned by SARS-CoV-2 infection, sex, and age. This has clinical relevance due to the availability of drugs such as Caplacizumab, which inhibits vWF–platelet interaction, and Crizanlizumab, which, by inhibiting P-selectin binding to its ligands, prevents leukocyte recruitment and platelet aggregation at the site of vascular damage.

## 1. Introduction

Congenital Thrombotic Thrombocytopenic Purpura (cTTP) is a severe autosomal recessive disorder due to *ADAMTS13* (a disintegrin-like and metalloprotease with thrombospondin type 1 motif, 13) rare variants and is characterized by uncleaved ultra-large vWF and thrombotic microangiopathy, frequently triggered by infections. Common polymorphisms in the same enzyme are reported to be involved in thrombosis [1,2]. It is well-recognized that coagulation abnormalities with an increased risk of thrombosis are one of the complications of severe Coronavirus disease 2019 (COVID-19) disease, accompanied by a high level of IL-6 and D-dimer often together with a reduction in platelets [3,4,5,6]. *ADAMTS13* gene encodes for a plasma glycoprotein with protease activity that plays a fundamental role in platelet adhesion and aggregation on vascular lesions, and the reduced activity of ADAMTS13 is already reported to be associated with a severe COVID-19 outcome [7]. Moreover, *ADAMTS13* protein production is positively induced by estrogen, and this reflects the greater penetrance of acquired or congenital TTP in middle aged females (over 50 years), whose estrogen levels start to decrease in relation to males [8].

In our previous work, we modeled COVID-19 by the use of Artificial Intelligence (AI), and we identified variants related to COVID-19 severity [9]. Here, we specifically explore the role of one key genetic player: ultra-rare variants in *ADAMTS13*.

## 2. Materials and Methods

### 2.1. GEN-COVID Cohort

A cohort of 2988 SARS-CoV-2-positive subjects, collected within the GEN-COVID Multicenter study, was used in this work, including 48 from the Netherlands ConCOVID cohort. Among the 2988 subjects, 1781 were males and 1207 were females. The majority (2808 subjects, 94%) were European; the remaining 180 (6%) subjects were of African, Asian, American, and Hispanic ethnicity.

### 2.2. Whole Exome Sequencing Analysis (WES)

WES, with at least 97% coverage at 20×, was performed using the NovaSeq6000 System (Illumina, San Diego, CA, USA) as previously described [10]. WES data were represented in a binary mode on a gene-by-gene basis [9,10,11].

### 2.3. Phenotype Definition Adjusting by Age

An Ordered Logistic Regression (OLR) Model was applied, separately for males and females, using age to predict the clinical grading according to the WHO Outcome scale [12]. Each patient had a clinical classification equal to 0 (asymptomatic cases) if the actual patient grading was below the one predicted by the OLR or 1 (severely affected cases) if the grading was above the OLR prediction. The patients with a predicted gradient equal to the actual gradient were excluded from further analyses, by which we wanted to compare the “extreme ends” [9,10,11]. After the adjustment in females, there were 486 subjects with OLR equal to 1; 289 subjects with ORL equal to 0; and 432 subjects with a predicted gradient equal to the WHO gradient. In males, there were 672 subjects with OLR equal to 1; 552 subjects with ORL equal to 0; and 557 subjects with a predicted gradient equal to the WHO gradient.

### 2.4. ADAMTS13 Assay

The TECHNOZYM^®^ ADAMTS-13 Activity Enzyme-Linked Immunosorbent Assay (ELISA) was used for determination of the ADAMTS13 activity using the venous-drawn, frozen, citrated (3.2% sodium citrate), platelet-poor plasma of studied patients. Blood samples should be collected with minimal stasis and processed rapidly to avoid cellular and plasma activation. The assay is a chromogenic and quantitative test performed on microplate readers capable of reading wavelengths of 450 nm. The measured value is reported as a percentage of normal-pooled plasma, which has been calibrated and defined as 100% activity [13].

### 2.5. Statistical Analysis

Statistical analysis was performed using R version 4.1.3 (10 March 2022) and RStudio (2022.02.0-443) software. A *p*-value < 0.05 was considered statistically significant.

## 3. Results

### 3.1. ADAMTS13 Ultra-Rare Variants Associate with Severity in COVID-19

Exome analysis of 2988 SARS-CoV-2-infected subjects of different severities, belonging to GEN-COVID cohort, stratified by sex and adjusted by age, shows an association between *ADAMTS13* ultra-rare variants (Minor Allele Frequency < 0.001) and severity in female patients with an OR = 3.32 (95% CI 1.37 to 8.05; *p*-value = 4.9 × 10^–3^) (Table 1a). No significant association was found in male patients (*p*-value = 0.252) (Table 1b). The adjustment by age was performed as explained in the paragraph “Phenotype definition adjusting by age” of the Section 2.

### 3.2. Characterization of Ultra-Rare Variants

One of several heterozygous *ADAMTS13* ultra-rare variants (Table 2), classified as either VUS or pathogenic, was identified in 124 SARS-CoV-2-infected patients (4.2%), including 49 females (39.5%) and 75 males (60.5%). Among these 124 subjects, 110 were of European ethnicity, and the remaining subjects were of African, Asian, and Hispanic ethnicities.

Most of the subjects (106) had severe COVID-19 disease requiring hospitalization (85.5%). The remaining 18 subjects (14.5%) were not-hospitalized patients (Table 2). The majority of not-hospitalized patients (14 subjects) were either females under 50 years or males over 50 years of age. For the females under 50, a protective role of estrogen, which increases *ADAMTS13* transcript, can be envisaged. Among the hospitalized patients, there were also three females younger than 10 months, as described below (Table 2).

### 3.3. Characterization of ADAMTS13 Activity of Ultra-Rare Variants

Eleven subjects (6 cases with *ADAMTS13* mutations and 5 controls without mutations) had ad hoc blood drawn and successful ADAMTS13 activity assessed after SARS-CoV-2 infection. The ADAMTS13 assay results were (median (min–max)) 61% (48–84) and 85% (71–106) for Cases and Controls, respectively. Carriers of ultra-rare variants show a significant reduction of ADAMTS13 activity, *p*-value = 0.017 (Wilcoxon test), as expected for heterozygous subjects. The box plot (Figure 1) shows the distribution of the two groups.

Among heterozygous subjects, there is a large variability in the percentage of activity, likely due to different effects of each mutation and to additional genetic factors modulating the activity. The mutation c.2915G > A, p.R972Q has 64% of activity (normal value of up to 150%); the mutation c.2111G > A, p.R704H has 48% of activity; the mutation c.2854C > T, p.P952S has 63.8% of activity (mean of 4 subjects with a range of 49–85).

There is no data about low levels of ADAMTS13 in the other carriers, whose ADAMTS13 activity has not been measured.

### 3.4. Laboratory Values in Heterozygous Subjects

During hospitalization, both males and females with heterozygous *ADAMTS13* variants have a tendency for hyper-inflammation (CRP mean 39, *p *= 0.005), higher D-dimer (mean 3024, *p* = 0.03), platelets consumption (platelet count mean 180, *p* = 0.07) and hemolysis (LDH mean 444, *p* = 0.009) (Table 3 and Appendix A).

The correlation is sustained mainly by females ≥50 years (CRP mean 55, *p* = 0.005; LDH mean 506, *p* = 0.006933) and males <50 years (platelet mean 153, *p* = 0.052) (Table 3 and Appendix A). No significant correlation was observed between fibrinogen levels and carriers of ultra-rare variants (Table 3 and Appendix A).

### 3.5. Autosomal Dominant Disorder Conditioned by SARS-CoV-2 Infection, Sex and Age

Complete clinical and molecular data were available for two families (Figure 2).

Data of segregation analysis were able to demonstrate that the disorder segregates as autosomal dominant disorders conditioned by SARS-CoV-2 infection, sex, and age (Figure 2). In the first family, the 66-year-old female who required oxygen support transmitted the mutation to the 34-year-old son who required CPAP treatment. In the second family, the 73-year-old female treated by oxygen support transmitted the mutation to the 40-year-old daughter who was oligosymptomatic, likely due to the relatively young age; her sister, the 76-year-old without the mutation, was oligosymptomatic.

### 3.6. Pediatric Cases

Among the 127 patients with *ADAMTS13* variants, three pediatric cases required hospitalization. Clinical and molecular characteristics are detailed below.

Case 1 (female) is the second child of a non-consanguineous couple of Filipino origin. Her pathological anamnesis is negative. The patient contracted SARS-CoV-2 infection when she was nine months old and her parents also turned out to be positive, but neither of them needed hospitalization, while her sister was negative. She had fever, cough, rhinitis, diuresis contraction and diarrhea (she tested negative for adenovirus and rotavirus); she did not have respiratory distress and her chest/lung ultrasound showed a B pattern. Among her blood tests, to be noticed: SGOT (serum glutamic oxaloacetic transaminase) 105 UI/L and SGPT (serum glutamic-pyruvic transaminase) 46 UI/l (maximum values during hospitalization) and D-dimer 1096 ng/mL. She has c.1016C > G, p.T339R *ADAMTS13* heterozygous variant (MAF ExAC_NFE 0.0004; MAF ExAC_SAS 0.0012).

Case 1 (female) is the second child of a non-consanguineous couple of Filipino origin. Her pathological anamnesis is negative. The patient contracted SARS-CoV-2 infection when she was nine months old and her parents also turned out to be positive, but neither of them needed hospitalization, while her sister was negative. She had fever, cough, rhinitis, diuresis contraction and diarrhea (she tested negative for adenovirus and rotavirus); she did not have respiratory distress and her chest/lung ultrasound showed a B pattern. Among her blood tests, to be noticed: SGOT (serum glutamic oxaloacetic transaminase) 105 UI/L and SGPT (serum glutamic-pyruvic transaminase) 46 UI/l (maximum values during hospitalization) and D-dimer 1096 ng/mL. She has c.1016C > G, p.T339R *ADAMTS13* heterozygous variant (MAF ExAC_NFE 0.0004; MAF ExAC_SAS 0.0012).

Case 2 (female) is the second child of a non-consanguineous European couple (Ukrainian mother, Italian father). She was born with a pulmonary CCAM (congenital cystic adenomatoid malformation) and a patent foramen ovale with a left-right shunt. The patient contracted SARS-CoV-2 infection when she was one month old and her sister and parents turned out to be positive but information on their clinical outcome is not available. She had fever, cough, rhinitis, diuresis contraction and diarrhea (she tested negative for adenovirus and rotavirus); she had slight respiratory distress, which did not require oxygen therapy, and her chest/lung ultrasound was negative. Among her blood tests, to be noticed: SGOT 61 UI/L and SGPT 36 UI/L at admission. She has c.1423C > T, p.P475S *ADAMTS13* heterozygous variant (MAF ExAC_NFE 0.0006).

Case 3 (female) is the fourth child of a non-consanguineous couple of Nigerian origin. She was diagnosed with Rubinstein-Taybi syndrome (22q13.1q13.2 deletion) when she was a newborn. The patient contracted SARS-CoV-2 infection when she was seven months old and her mother turned out to be positive and she was not hospitalized. She had fever, cough, dyspnea, tachycardia, and respiratory distress (with oxygen saturation 88–89 and resuscitation); she also had a rhinovirus infection and part of her clinical picture is due to her genetic condition: in fact, she has a congenital heart defect. Her chest/lung ultrasound showed a B pattern. Among her blood tests, to be noticed: SGOT 52 UI/L and SGPT 71 UI/L at admission, which were normal when the child was discharged. She has c.2494G > A, p.V832M *ADAMTS13* heterozygous variant (MAF ExAC_NFE 0.000017, MAF ExAC_AFR 0.013).

## 4. Discussion

Patients with severe COVID-19 can develop a wide range of complications, the most common of which is thrombosis of the large vessels. Thrombotic microangiopathy (TMA), whose pathophysiology is mostly due to endothelial dysfunction, has only been described in a few of these patients. TMAs include (i) congenital Thrombotic Thrombocytopenic Purpura (cTTP) characterized by no evidence of anti-ADAMTS-13 IgG antibodies and severe deficiency of *ADAMTS13* activity and (ii) autoimmune TTP characterized by the presence of anti-ADAMTS-13 IgG antibodies. In primary autoimmune disease, no clear cause is identified, and instead in secondary autoimmune TTP a defined disorder or trigger can be identified, such as a viral infection [14,15]. The literature, recently reviewed by Singh B. et al. [16], contains few case reports of secondary autoimmune TTP in the course of COVID-19. In the descriptions of clinical cases, it is difficult to distinguish whether there is a cause-and-effect relationship between COVID-19 and TTP or whether SARS-CoV-2 infection is just present at the time of TTP diagnosis.

All cases of TTP are due to reduced activity of ADAMTS13, the enzyme involved in the cleavage of ultra-large von Willebrand factor (vWF) multimers into smaller, less procoagulant multimers. The congenital or inherited form of TTP has autosomal recessive inheritance, a prevalence of 0.5–2 cases per million [17], and accounts for 2–10% of all TTP cases reported in international registries [18]. The diagnosis of secondary autoimmune TTP is possible in the presence of microangiopathic hemolytic anemia, thrombocytopenia, ADAMTS13 activity <10%, and demonstration of an anti-ADAMTS13 inhibitor [19]. TTP heterozygous, i.e., carriers of one mutated allele only, are reported to be healthy. Here, we show evidence that heterozygous subjects are at risk for severe COVID-19 through a micro-thrombotic mechanism. Furthermore, the disease segregates in families as an autosomal dominant disorder, conditioned by SARS-CoV-2 infection, sex, and age. It is also known that the TTP-recessive disease is more penetrant in females. Females have a lower basal level of ADAMTS13 than males. However, estrogens have the power to induce protein production. Indeed, we expect females from the puberal period until ovarian failure to be protected by the action of estrogens [8]. In line with this, we have identified that heterozygous females over 50 are at more risk. On the other hand, we have reported pediatric cases as well (all females), which also miss the beneficial effect of estrogens. In the other sex (male), the period with less estrogens is that from puberty to andropause and indeed, as shown by laboratory value, the tendency towards microangiopathy is more evident in males under 50.

The penetrance of the thrombotic disease triggered by SARS-CoV-2 infection in heterozygous *ADAMTS13* subjects is incomplete. Other factors that may contribute to the imbalance in the vWF antigen (VWF:Ag):/ADAMTS13 ratio are age, as vWF levels increase with age [20], and the patient’s membership in a blood group other than 0, resulting in baseline vWF:Ag levels 25–30% higher than in group 0 patients [21]. Furthermore, common polymorphisms in thrombotic microangiopathy-associated genes such as the rs2230199 in *C3*, the rs800292 in *CFH* (26 patients), and the rs2301612 missense mutation (448E) in the gene itself *ADAMTS13*, reported in 60 patients with moderate-to-severe COVID-19 studied by Graviilaki E. et al., may contribute to penetrance modulation [22]. From a multistep pathogenetic perspective of TMA, the procoagulant environment that originates during SARS-CoV-2 infection could precipitate the clinical manifestation of TMA in patients with genetic variants of *ADAMTS13*. There is indeed a direct virus-induced endothelial damage and secondary inflammatory status to cytokine storms [23], which result in the release of vWF from endothelial storage sites and a further reduction in ADAMTS13 activity, creating an imbalance in the vWF: Ag/ADAMTS13 activity ratio [24,25,26,27].

## 5. Conclusions

In conclusion, data from the large multicenter GEN-COVID study allow us to define the prevalence of *ADAMTS13* mutations in a SARS-CoV-2-positive population and to establish the severity of COVID-19 pathology in patients carrying the mutation. This finding has clinical relevance due to the availability of drugs such as Caplacizumab or Crizanlizumab that could be suggested to patients with *ADAMTS13* variants exhibiting decreased enzymatic activity. Caplacizumab, an anti-vWF bivalent single-domain nanobody, inhibits vWF–platelet interaction and is already used to treat acquired thrombotic thrombocytopenic purpura. Besides, Crizanlizumab is a monoclonal antibody that prevents leukocyte recruitment and platelet aggregation at the site of vascular damage by inhibiting P-selectin binding to its ligands. These two drugs are likely to replace the reduced activity of the metalloproteinase due to certain mutations and therefore they could also be useful in decreasing hyper-inflammation signs in heterozygous *ADAMTS13* patients.

## Figures and Tables

**Figure 1 viruses-14-01185-f001:**
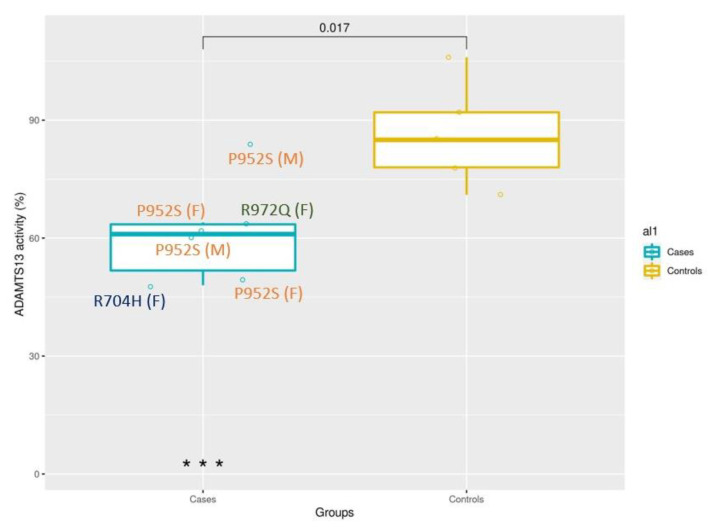
Heterozygous *ADAMTS13* ultra-rare variants are related to a reduction of protein detection. Box plot of patients with one ultra-rare variant (6 cases) and patients without ultra-rare variants (5 controls). The presence of ultra-rare variants is associated with a reduction of ADAMTS13 activity (*p*-value = 0.017 at Mann–Whitney U test).

**Figure 2 viruses-14-01185-f002:**
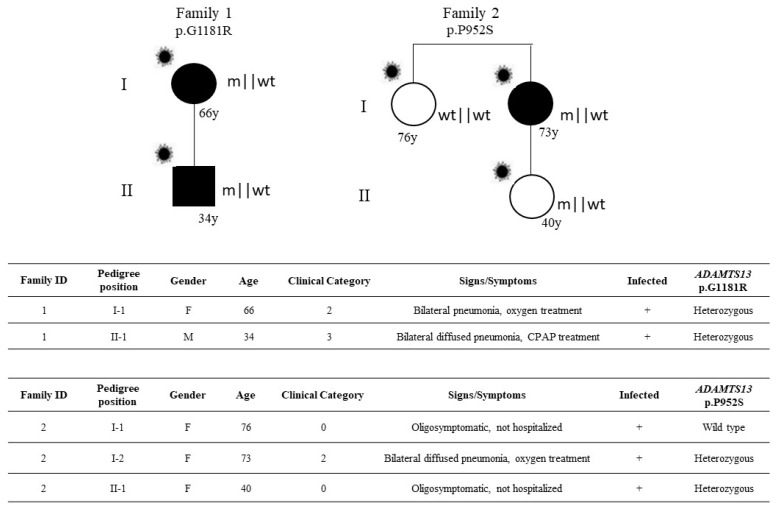
Segregation analysis. Pedigree (**upper panel**) and respective segregation of *ADAMTS13* variant and COVID-19 status (**lower panel**) are shown. Squares represent male family members; circles represent females. A virus cartoon close to the individual symbol indicates individuals infected by SARS-CoV-2 (
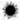
). The inheritance pattern appears that of an autosomal dominant disorder conditioned by SARS-CoV-2 infection, sex, and age.

**Table 1 viruses-14-01185-t001:** (**a**) *ADAMTS13* ultra-rare variants correlation with COVID-19 severity in female cohort. (**b**) *ADAMTS13* ultra-rare variants correlation with COVID-19 severity in male cohort.

(**a**)			
**Phenotype**	**Ultra-Rare Variants**	**Wild Type**	**Total**
**Severe**	32	454	486
**Not severe**	6	283	289
**Total**	38	737	775 (Grand Total)
OR = 3.32 (95% CI 1.37 to 8.05); *p*-value = 4.9 × 10^−3^			
(**b**)			
**Phenotype**	**Ultra-Rare Variants**	**Wild Type**	**Total**
**Severe**	23	649	672
**Not severe**	26	526	552
**Total**	49	1175	1224 (Grand Total)
*p*-value = 0.252888			

Note: The correlation was obtained by chi-square test; *p*-value (severe vs not severe cases), significant at *p* < 0.05. Severe = adjusted by age category 1; Not severe = adjusted by age category 0.

**Table 2 viruses-14-01185-t002:** *ADAMTS13* heterozygous mutations in the entire cohort of SARS-CoV-2 positive subjects.

Nucleotide Change	Amino Acid Change	dbSNP	CADD	ExAC_NFE	Tot. n. Patients	Sex (n.)	Age Range	Hospitalized (Not Hospitalized)	Category †
c.11G > A	p.R4H	rs370406676	5.2	0.0001	1	F	56	1	2
c.220C > T	p.R74W	n.a.	22	0.000008	1	M	39	1	1
c.241C > T	p.H81Y	rs148644959	23	n.a	4	M (2)	38–60	2	1
						F (2)	60–68	2	1
c.353C > T	p.P118L	rs587698109	19.3	0.000008	1	F	59	1	3
c.559G > C	p.D187H	rs148312697	25.6	0.0006	7	M(5)	41–76	4	3–2
							54	(1)	0
						F (2)	50–82	2	3–2
c.649G > A	p.D217N	rs782305581	29.4	0.00003	2	M(2)	75	1	3
							34	(1)	0
c.703G > T	p.D235Y	n.a.	33	0.00004	1	F	45	1	2
c.722G > C	p.G241A	n.a.	8.1	n.a.	v1	F	30	(1)	0
c.742G > A	p.V248M	n.a.	25.1	0.00004	2	M	49–51	2	3–2
c.953A > G	p.K318R	n.a.	0.006	n.a.	1	F	42	(1)	0
c.1016C > G	p.T339R	rs149517360	22.8	0.0004 ^#^	6	M	40	1	3
						F (5)	9 months-66	4	2–1
							23	(1)	0
c.1084G > A	p.V362M	rs781924046	0.21	0.00001523	1	F	46	(1)	0
c.1117_1121del	p.S373Gfs*15	n.a	n.a	n.a	1	M	39	1	1
c.1157G > A	p.R386H	rs151048660	11.6	0.0003	10	F (5)	46–82	4	4–3
							49	(1)	0
						M (4)	42–73	4	3–2
c.1178G > A	p.R393Q	rs140937290	12.5	0.000017	1	M	68	1	1
c.1226G > A	p.R409Q	n.a	35	n.a.	1	M	48	1	1
c.1261C > T	p.R421C	rs145825553	33	0.0008	6	M (6)	55–81	4	3–1
							52–65	(2)	0
c.1291G > A	p.E431Q	rs781915989	25.8	0.000018	2	M	47	(1)	0
						F	75	(1)	0
c.1336A > G	p.M446V	rs782733359	16.1	0.000022	1	F	70	1	3
c.1423C > T	p.P475S	rs11575933	4.5	0.0006 ^§^	3	M (2)	33	1	1
							63	(1)	0
						F	1 month	1	1
c.1463G > A	p.R488Q	rs147201977	22.6	n.a.	1	M	72	1	5
c.1486A > G	p.M496V	rs782574335	0.001	0.00004	1	F	44	1	1
c.1492C > A	p.R498S	n.a.	32	n.a.	1	F	93	1	2
c.1601G > A	p.G534D	rs782003053	26.6	0.00005	1	M	62	1	2
c.1700C > T	p.A567V	rs782272645	27.4	0.00007	1	M	79	1	2
c.1729A > T	p.T577S	n.a.	8.6	n.a.	1	F	33	1	1
c.1753A > G	p.I585V	n.a.	0.001	n.a.	1	M	82	1	3
c.1808A > G	p.Y603C	rs867154790	24.2	n.a.	1	M	52	1	1
c.1906C > T	p.R636W	rs201704847	24.7	0.000008	2	M	57–62	2	3–2
c.1931G > A	p.R644H	rs782184721	0.011	0.00002	1	F	70	1	2
c.1976G > A	p.R659K	rs150764227	23.5	0.0003	1	M	57	1	2
c.2009G > A	p.R670H	rs149953167	10.7	0.0003	1	F	78	1	5
c.2011C > A	p.P671T	n.a.	22.1	n.a.	1	F	56	1	3
c.2038C > T	p.P680S	n.a.	24.3	n.a.	1	M	76	1	2
c.2099G > A	p.G700E	n.a.	31	n.a.	1	M	25	1	3
c.2111G > A	p.R704H	rs782223605	23.7	0.0000008	1	F	34	1	3
c.2111G > T	p.R704L	n.a.	26.2	n.a.	1	M	68	1	2
c.2278G > A	p.G760S	rs782729939	22.8	0.00005	1	F	57	1	2
c.2282G > T	p.G761V	n.a.	25.3	n.a.	1	F	49	(1)	0
c.2288G > A	p.R763Q	rs781804540	16.8	0.000020	1	M	60	1	2
c.2351G > A	p.R784Q	rs377187626	4.4	n.a.	1	M	57	1	2
c.2420G > C	p.R807T	n.a.	23.5	n.a.	1	M	71	1	2
c.2422C > T	p.W808R	n.a.	0.007	n.a.	1	M	50	(1)	0
c.2494G > A	p.V832M	rs34104386	18.5	0.000017 ^	2	M	28	1	1
						F	7 months	1	2
c.2519C > T	p.A840V	n.a.	0.3	n.a.	1	F	67	1	2
c.2545G > A	p.V849I	rs140639242	0.4	0.0002	1	M	72	1	5
c.2773A > G	p.R925G	rs782263547	4.1	0.000009	2	M	57–65	2	4–3
c.2814G > T	p.K938N	n.a.	25.7	n.a.	2	M	57–72	2	4–2
c.2824C > T	p.R942W	rs929435102	27.7	0.000009	2	M	61	(1)	0
						F	56	1	2
c.2828G > A	p.R943Q	rs782160285	2.6	0.00009	1	M	84	1	5
c.2854C > T	p.P952S	rs143568784	29.9	0.0003	5	M	68	1	2
						F (4)	67–85	3	2
							40	(1)	0
c.2915G > A	p.R972Q	rs139951127	5.4	0.0002	7	M (4)	37–78	3	3–2
							50	1	0
						F (3)	35–70	1	5–1
c.2978C > T	p.T993I	rs139808736	23.2	0.00006	1	M	57	1	3
c.3161delC	p.Cys1055Valfs*66	n.a.	n.a.	n.a.	1	F	54	1	2
c.3201T > A	p.C1067 *	n.a.	36	n.a.	1	M	72	1	3
c.3356C > T	p.P1119L	rs1044262941	36	0.000009	1	M	64	1	2
c.3463G > A	p.A1155T	n.a.	1.6	n.a	1	M	37	1	2
c.3541G > A	p.G1181R	rs192619276	1.5 *	0.000009 °	5	M (3)	34–74	3	3–1
						F (2)	57–66	2	3–2
c.3685G > A	p.V1229I	rs587643681	2.5	0.00001769	1	M	56	1	1
c.3694A > T	p.S1232C	n.a.	23.6	0.00001769	1	M	50	1	1
c.3713C > T	p.A1238V	rs587697598	13.9	0.00007986	1	F	63	1	3
c.3718G T	p.D1240Y	n.a.	24.9	n.a.	1	M	60	1	2
c.3722T > C	p.M1241T	rs1057522240	0.002	0.000008	1	F	46	1	1
c.3740G > A	p.R1247Q	rs782197792	27.2	0.00004	1	M	34	(1)	0
c.3826G > A	p.G1276R	rs144808448	0.493	0.00003	1	M	62	1	5
c.3853C > T	p.R1285W	rs370157837	27.6	0.00002264	1	M	68	1	4
c.3956C > T	p.T1319M	rs375824927	8.19	n.a.	1	F	84	1	2
c.3962A > T	p.N1321I	rs200645384	1.248	0.00006	1	M	73	1	2
c.4007G > A	p.R1336Q	rs782213090	23.8	0.000008	1	M	53	1	2
c.4012G > A	p.A1338T	rs782401854	27	0.000008	1	M	60	1	3
c.4141T > G	p.S1381A	n.a.	25.6	n.a.	1	F	29	(1)	0
c.4262_4271del	p.G1423Efs*6	n.a.	n.a.	n.a.	1	M	48	1	3

Note: * mutation already reported as pathogenic in the Clinvar database (https://www.ncbi.nlm.nih.gov/clinvar/ (accessed on 21 April 2022)). # ExAC_SAS = 0.0012; § ExAC_EAS = 0.015, ExAC_AMR = 0.02; ^ ExAC_AFR = 0.013; ^°^ ExAC_EAS = 0.022; † Clinical category: 5, Deceased; 4, Hospitalized and intubated; 3, Hospitalized and CPAP-BiPAP and high-flows oxygen treated; 2, Hospitalized and treated with conventional oxygen support only; 1, Hospitalized without respiratory support; 0, Not hospitalized oligo/asymptomatic individuals. CADD, Combined Annotation Dependent Depletion; ExAC_NFE, Non-Finnish European minor allele frequency.

**Table 3 viruses-14-01185-t003:** Correlations between *ADAMTS13* ultra-rare variants and laboratory values in hospitalized patients.

**CRP M and F cases**			**CRP M < 50 y cases**			**CRP F ≥ 50 y cases**		
Ultra-rare variants	Mean	Count	Ultra-rare variants	Mean	Count	Ultra-rare variants	Mean	Count
yes	39	64	yes	24.5	12	yes	55.1	21
no	28.5	1491	no	18.7	163	no	27.1	443
*p*-value = 0.0005166			*p*-value = 0.1116			*p*-value = 0.0001896		
**Fibrinogen M and F cases**			**Fibrinogen M < 50 y cases**			**Fibrinogen F ≥ 50 y cases**		
Ultra-rare variants	Mean	Count	Ultra-rare variants	Mean	Count	Ultra-rare variants	Mean	Count
yes	488	25	yes	446	6	yes	502	8
no	502	806	no	499	67	no	503	234
*p*-value = 0.8843			*p*-value = 0.6227			*p*-value = 0.9406		
**D-Dimer M and F cases**			**D-Dimer M < 50 y cases**			**D-Dimer F ≥ 50 y cases**		
Ultra-rare variants	Mean	Count	Ultra-rare variants	Mean	Count	Ultra-rare variants	Mean	Count
yes	3024	58	yes	4016	8	yes	4127	18
no	2788	1446	no	1908	148	no	2711	441
*p*-value = 0.03431			*p*-value = 0.1889			*p*-value = 0.47		
**Platelets M and F cases**			**Platelets M <50 y cases**			**Platelets F ≥ 50 y cases**		
Ultra-rare variants	Mean	Count	Ultra-rare variants	Mean	Count	Ultra-rare variants	Mean	Count
yes	183	63	yes	153	11	yes	182	17
no	315	1509	no	221	184	no	574	451
*p*-value = 0.07816			*p-*value = 0.05286			*p*-value = 0.2213		
**LDH M and F cases**			**LDH M <50 y cases**			**LDH F ≥ 50 y cases**		
Ultra-rare variants	Mean	Count	Ultra-rare variants	Mean	Count	Ultra-rare variants	Mean	Count
yes	444	49	yes	513	8	yes	506	16
no	395	1313	no	392	142	no	374	389
*p*-value = 0.009494			*p*-value = 0.05761			*p*-value = 0.006933		

Note: CRP (C-reactive Protein) (mg/dL) highest value among all those collected during hospitalization; normal value <0.5 mg/dL; Fibrinogen (mg/dL) lowest value among all those collected during hospitalization; n.v. 200–400 mg/dL; D-Dimer (ng/mL) highest value among all those collected during hospitalization; n.v. <500 ng/mL; Platelets (10^3^/mmc) lowest value among all those collected during hospitalization; n.v. 150–450 × 10^3^/mmc; LDH (Lactate dehydrogenase) (UI/L) highest value among all those collected during hospitalization; n.v. 135–225 UI/L (male (M)); 135–214 UI/L (female (F)). For the correlations, the Mann-Whitney U test was performed; *p*-value is significant at *p* < 0.05. Correlations were performed in hospitalized patients using both sexes (M and F cases), males under 50 years of age (M < 50y cases) and females over 50 years of age (F **≥ **50y cases). Complete laboratory values correlation were included in Appendix A.

## Data Availability

The data are available for sharing through the COVID-19 dedicated section (http://nigdb.cineca.it (accessed on 1 March 2022)), within the Network for Italian Genome (http://www.nig.cineca.it (accessed on 1 March 2022)). The data and samples referenced here are housed in the GEN-COVID Patient Registry and the GEN-COVID Biobank and are available for consultation. You may contact the corresponding author, Prof. Alessandra Renieri (e-mail: alessandra.renieri@unisi.it).

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
