# Peer review of "Carriers of ADAMTS13 Rare Variants Are at High Risk of Life-Threatening COVID-19"

_viruses, 2022, doi:10.3390/v14061185_

Round 1
Reviewer 1 Report
This manuscript reports that ADAMTS13 mutations are more frequent in patients with severe COVID-19 than in a control group. Since these mutations may predispose to micro-thrombi, as in TTP, and since thrombosis may be part of severe COVID-19, this finding is interesting.
My main criticism is that I found the data greatly over-interpreted and presented in a somewhat misleading manner throughout the manuscript, in the abstract and even in the title. First, I find the term ‘TTP carriers’ rather poor terminology to say that heterozygotes for ADAMTS13 may or may not develop TTP. Second and most important, in this paper there is no single case described that even remotely qualifies for a diagnosis of TTP.
Table 1 shows only data in females: why not show also the data in males? Some of the numbers do not add up. 29 + 6 = 35, rather than 38; 35 + 762 = 797, rather than 687. Also, in the text there are 49 females with ‘ultra-rare’ variants, but in Table 1 they are 35.
What are ‘wild-type variants’ (plural)? Wild-type, be definition, should be only one.
Caucasian is an obsolete ethnic classification from the times when Caucasian was regarded as a ‘race’: I think the Authors mean European.
Fig. 1. If I understand correctly, the data are from 6 cases with ADAMTS13 mutations and 10 cases without mutations (this should have been clearly stated, rather than forcing the reader to work it out). Only 3 mutations are involved, out of the many in Table 2: therefore it should be clearly stated that we don’t know how many of the heterozygotes had low ADAMTS13 levels.
Microangiopathy is a histopathological feature, and it can be inferred from red cell changes on the peripheral blood smear: here neither are shown, and therefore this sub-heading is untenable, not to say wrong.
Table 3. Most of the differences are not statistically significant; and when they are, it is essentially thanks to the large number of (non-severe) control cases. The claim for platelet ‘consumption’, when the mean platelet count is 180, i.e. entirely normal, is incredible. The claim for ‘hyper-inflammation’, when the CRP is 39 versus 28.5, and the D-dimer is 3024 versus 2788, is over-stated to say the least.
Fig. 2. In family 2 it is possible that young age has played a role, but this pedigree cannot be claimed as supporting autosomal dominant inheritance: it shows the opposite!
The three cases in children are clinically interesting, but no evidence is given for microangiopathic haemolytic anaemia, let alone TTP.
Most of the discussion about pathogenesis of TTP is irrelevant. In the last paragraph, when the Authors say that “The penetrance of disease is incomplete”, they seem to imply that there is a specific disease that occurs when a heterozygote for a ADAMTS13 pathogenic mutation is infected by SARS-CoV2. The data do not remotely support this.
In my view the Authors ought to thoroughly re-write the paper, including title and abstract. They have not shown that heterozygotes for ADAMTS13 mutations can develop TTP. What they have shown is that in females ‘ultra-rare’ ADAMTS13 mutations are less rare in severe COVID-19 (7.3%) than in non-severe COVID-19 (2%).
Author Response
Reviewer #1: comments
This manuscript reports that ADAMTS13 mutations are more frequent in patients with severe COVID-19 than in a control group. Since these mutations may predispose to micro-thrombi, as in TTP, and since thrombosis may be part of severe COVID-19, this finding is interesting.
My main criticism is that I found the data greatly over-interpreted and presented in a somewhat misleading manner throughout the manuscript, in the abstract and even in the title. First, I find the term ‘TTP carriers’ rather poor terminology to say that heterozygotes for ADAMTS13 may or may not develop TTP. Second and most important, in this paper there is no single case described that even remotely qualifies for a diagnosis of TTP.
We thank the reviewer for this comment and accordingly we changed the title as follows: “Carriers of ADAMTS13 rare variants are at high risk of life-threatening COVID-19”.
We also revised the abstract and the manuscript according to the reviewer’s comment.
Concerning the absence of diagnosis of TTP in described cases, in this manuscript we report the association between ADAMTS13 heterozygous mutations and severe COVID-19 through an increased risk of thrombosis which is one of the well-known complications of severe COVID-19. The diagnosis of TTP is confirmed by the presence of homozygous or compound heterozygous ADAMTS13 mutation and in this paper, we described only heterozygous subjects as subjects at higher risk of severe COVD-19.
- Table 1 shows only data in females: why not show also the data in males? Some of the numbers do not add up. 29 + 6 = 35, rather than 38; 35 + 762 = 797, rather than 687.
We have updated the numbers in Table 1. We also added the data in males as table 1b to show the difference in significance between males and females.
- Also, in the text there are 49 females with ‘ultra-rare’ variants, but in Table 1 they are 35.
We updated the numbers in Table 1a and there are 38 females from the GEN-COVID cohort bearing ultra-rare variants. Later in the text we mention that there are 49 females with ultra-rare variants and this is because here we have included females of all clinical grading according to the WHO Outcome scale. In Table 1a, the chi square is calculated only on the extreme ends of adj by age classification (category 1: severe) versus 0: not severe) as described in the method section “Phenotype definition adjusting by age”.
- What are ‘wild-type variants’ (plural)? Wild-type, be definition, should be only one.
We agree that the term ‘wild-type variants’ is not correct. We removed “variants” and left just “Wild type” referring to the phenotype.
- Caucasian is an obsolete ethnic classification from the times when Caucasian was regarded as a ‘race’: I think the Authors mean European.
Yes, by Caucasian ethnicity we mean European, and we changed it in the text.
- Fig. 1. If I understand correctly, the data are from 6 cases with ADAMTS13 mutations and 10 cases without mutations (this should have been clearly stated, rather than forcing the reader to work it out). Only 3 mutations are involved, out of the many in Table 2: therefore it should be clearly stated that we don’t know how many of the heterozygotes had low ADAMTS13 levels.
The data depicted in Fig. 1 are from 6 cases and 5 controls. Out of sixteen subjects mentioned in the text, 11 had ADAMTS13 mutations and 5 did not. From the cases, we removed the outliers and the values of two subjects bearing a not pathogenic ultra-rare mutation in ADAMTS13 and then performed the statistical analysis shown in Figure 1. We added more information in the text regarding Figure 1 analysis.
As the reviewer suggested, we added a statement at the end of the paragraph to make it clear that we have no data regarding ADAMTS13 activity in the other carriers whose protein level was not measured.
- Microangiopathy is a histopathological feature, and it can be inferred from red cell changes on the peripheral blood smear: here neither are shown, and therefore this sub-heading is untenable, not to say wrong.
According to the reviewer's suggestion, we changed the subheading into “Laboratory values in heterozygous subjects”.
- Table 3. Most of the differences are not statistically significant; and when they are, it is essentially thanks to the large number of (non-severe) control cases. The claim for platelet ‘consumption’, when the mean platelet count is 180, i.e. entirely normal, is incredible. The claim for ‘hyper-inflammation’, when the CRP is 39 versus 28.5, and the D-dimer is 3024 versus 2788, is over-stated to say the least.
We agree with the reviewer that not all differences are statistically significant and further analysis are needed in order to confirm the results in a larger cohort of both cases (severe) and controls (not severe) subjects. According to the reviewer's suggestion we have rephrased the paragraph.
- Fig. 2. In family 2 it is possible that young age has played a role, but this pedigree cannot be claimed as supporting autosomal dominant inheritance: it shows the opposite!
We thank the reviewer for this comment. As described in the text, the autosomal dominant inheritance in subjects with heterozygous ADAMTS13 variants and SARS-CoV-2 infection appears to be conditioned by sex and age. The penetrance is incomplete, and in the 40-year-old female, we have to consider the presence of other genetic factors that could mitigate the phenotype.
- The three cases in children are clinically interesting, but no evidence is given for microangiopathic haemolytic anaemia, let alone TTP.
We agree with the reviewer that the three pediatric cases have no TMA or TTP, but they are clinically interesting. In fact, it is interesting to note that 3 of 127 (2.4%) cases with heterozygous ADAMTS13 variants were pediatric and that they are all females requiring hospitalization despite their very young age. Moreover, although we cannot speak of TMA or TTP, these cases showed a tendency to have signs of hyper inflammation with high D-dimer and SGOT/SGPT values.
- Most of the discussion about pathogenesis of TTP is irrelevant. In the last paragraph, when the Authors say that “The penetrance of disease is incomplete”, they seem to imply that there is a specific disease that occurs when a heterozygote for a ADAMTS13 pathogenic mutation is infected by SARS-CoV2. The data do not remotely support this.
We thank the reviewer for this comment and we have changed and better rephrased the discussion accordingly.
- In my view the Authors ought to thoroughly re-write the paper, including title and abstract. They have not shown that heterozygotes for ADAMTS13 mutations can develop TTP. What they have shown is that in females ‘ultra-rare’ ADAMTS13 mutations are less rare in severe COVID-19 (7.3%) than in non-severe COVID-19 (2%).
With this scientific work, we are trying to describe how heterozygous ultra-rare variants in ADAMTS13 identified in our cohort lead to a rare severe micro-thrombotic form of COVID-19. To support this statement, we provided all the available information, laboratory values and family cases. It was not our intention to show that heterozygotes for ADAMTS13 mutations can develop TTP. We made many changes in the text to clarify the overall message of this paper.
Reviewer 2 Report
The authors here report that ultra-rare variants in ADAMTS13 gene may lead to a rare severe micro-thrombotic form of COVID-19, which segregates in families as an autosomal dominant disorder. The manuscript topic is very interesting and findings are novel. However, the following revisions are needed:
- Title: "carriers of thrombotic thrombocytopenic purpura" should be changed to "carriers of ADAMTS13 gene variants"
- Abstract and introduction: this statement is not correct: "Thrombotic Thrombocytopenic Purpura is a severe autosomal recessive disorder characterized by uncleaved ultra-large vWF and thrombotic microangiopathy triggered by infections". Congenital TTP is a form of TTP and not necessarily triggered by infections.
- Introduction: This statement is not clear: “Moreover, ADAMTS13 protein production is positively induced by estrogen and this reflects the more penetrance of TTP in females with respect to males”.
- Tables (1,2,3): title and legends should be better explained
- Results: this statement is not clear: “over 50 years of age, for whom a protective role of estrogen, which increases ADAMTS13 transcript, can be envisaged”; the paragraph “Characterization of ADAMTS13 activity of ultra-rare variants” should be proof-read better; in the paragraph “Microangiopathy in carrier subjects” a statement concerning FBG and LDH should be added.
- Discussion: This statement is not correct: “Primary TMAs are the consequence of genetic deficits of one or more 83 factors as in the case of complement regulatory proteins in Atypical Hemolytic-Uremic 84 Syndrome (aHUS) or ADAMTS13 (A Disintegrin And Metalloproteinase with a 85 Thrombospondin type 1 motif, number 13) deficiency in Thrombotic Thrombocytopenic 86 Purpura (TTP). Secondary TMAs are, more often, the consequence of autoimmune disorders leading to the production of autoantibodies, as in the case of anti-ADAMTS13 in acquired TTP. ” Refer to the definitions of Scully et al, 2017 (Consensus on the standardization of terminology in thrombotic thrombocytopenic purpura and related thrombotic Microangiopathies)
- Discussion and conclusions: the relevance and the implications of the study should be explained, as well as the considerations concerning caplacizumab and crizanlizumab.
Author Response
Reviewer #2: comments
The authors here report that ultra-rare variants in ADAMTS13 gene may lead to a rare severe micro-thrombotic form of COVID-19, which segregates in families as an autosomal dominant disorder. The manuscript topic is very interesting and the findings are novel. However, the following revisions are needed:
- Title: "carriers of thrombotic thrombocytopenic purpura" should be changed to "carriers of ADAMTS13 gene variants"
We changed the title as follows: “Carriers of ADAMTS13 rare variants are at high risk of life-threatening COVID-19”.
- Abstract and introduction: this statement is not correct: "Thrombotic Thrombocytopenic Purpura is a severe autosomal recessive disorder characterized by uncleaved ultra-large vWF and thrombotic microangiopathy triggered by infections". Congenital TTP is a form of TTP and not necessarily triggered by infections.
We agree that TTP is not necessarily triggered by infections. We added “frequently” in the sentence since there is evidence in literature that multiple viral infections have been described to contribute to triggering TTP.
- Introduction: This statement is not clear: “Moreover, ADAMTS13 protein production is positively induced by estrogen and this reflects the more penetrance of TTP in females with respect to males”.
There is evidence that estrogen increases ADAMTS13 transcript, thus middle aged women, over 50 years, whose estrogen levels start to decrease might be more susceptible. We rephrased the sentence in the text.
- Tables (1,2,3): title and legends should be better explained
We Have better explained the title and legend of all tables.
- Results: this statement is not clear: “over 50 years of age, for whom a protective role of estrogen, which increases ADAMTS13 transcript, can be envisaged”; the paragraph “Characterization of ADAMTS13 activity of ultra-rare variants” should be proof-read better; in the paragraph “Microangiopathy in carrier subjects” a statement concerning FBG and LDH should be added.
The protective role of estrogen regards females under 50 years of age. Thus, we rewrote the second part of the sentence as a new sentence to make it clear. In the paragraph “Characterization of ADAMTS13 activity of ultra-rare variants” we added more details to better explain the performed analysis, and to make more clear what we are trying to communicate. We changed the title of the paragraph “Microangiopathy in carrier subjects” to “Laboratory values in heterozygous subjects”. We added the mean value of LDH and the p-value for both the total cohort of heterozygous subjects and for females over 50. Furthermore, we also included a statement regarding the correlation between FBG levels and the carriers as suggested.
- Discussion: This statement is not correct: “Primary TMAs are the consequence of genetic deficits of one or more 83 factors as in the case of complement regulatory proteins in Atypical Hemolytic-Uremic 84 Syndrome (aHUS) or ADAMTS13 (A Disintegrin And Metalloproteinase with a 85 Thrombospondin type 1 motif, number 13) deficiency in Thrombotic Thrombocytopenic 86 Purpura (TTP). Secondary TMAs are, more often, the consequence of autoimmune disorders leading to the production of autoantibodies, as in the case of anti-ADAMTS13 in acquired TTP. ” Refer to the definitions of Scully et al, 2017 (Consensus on the standardization of terminology in thrombotic thrombocytopenic purpura and related thrombotic Microangiopathies).
We have rephrased the paragraph according to the suggested reference. We added the suggested reference as number [15].
- Discussion and conclusions: the relevance and the implications of the study should be explained, as well as the considerations concerning caplacizumab and crizanlizumab.
In the “Conclusions” paragraph, we added more information about the mentioned drugs, and why these drugs should be used in patients bearing rare variants in ADAMTS13.
Round 2
Reviewer 1 Report
Abstract. The Authors have found an association: it is not correct to say that the heterozygous state for an ADAMTS13 mutation ‘leads’ to a severe form of COVID-19, since in many cases it does NOT lead to this.
Page 4. The phrase ‘One heterozygous ADMATS13 ultra-rare variant’ gives the impression that it is always the same. I think the Authors mean: “One of several ADMATS13 ultra-rare variants (see Table 2)….”
Fig. 1. The data and their presentation are unsatisfactory for several reasons. (1) Out of 124 patients with ultra-rare variants only 11 were tested; out of hundreds of ‘controls’ only 5 were tested. (2) Of the 11 'cases' we see data on only 6; I don’t know what it means that outliers were removed: in my view there is no justification for removing any data. (3) The control group is homogeneous in being presumably wild-type for ADAMTS13; the ‘cases’ group is heterogeneous, because it contains subjects who have different mutations. (4) Of the 'cases', 4 out of 6 have the same mutation, therefore they are not representative of the multitude of mutations listed in Table 2. Thus the data are over-interpreted, not to say misleading.
Fig. 2. The claim of autosomal dominant inheritance of severe COVID-19 based on 2 families (with only two generations and only one child in each), in one of which it has NOT occurred, borders on absurdity. As for segregation, one needs at least two children from a heterozygote, one inheriting the mutant gene and the disease, the other the wild type gene and no disease. This section, the corresponding statement in the abstract, and the corresponding paragraph in the discussion, need to be re-written.
Conclusion. At the moment caplacizumab is indicated for acquired thrombotic thrombocytopenic purpura in combination with plasma exchange and immunosuppressive therapy; crizanlizumab is indicated to reduce the frequency of vaso-occlusive crises in patients with sickle cell disease. The Authors are perfectly entitled to tentatively suggest that these drugs might have a role also in patients with COVID-19 who are heterozygous for certain ADAMTS13 mutations: but, in my view, in doing so they should be more cautious and a little more modest, especially since they have demonstrated in these patients only hyper-inflammation, not thrombophilia. The Authors are not entitled to say that these drugs “will… prevent thrombotic events”.
Overall, the Authors have found an interesting association between ADAMTS13 mutations and severe COVID-19. In my view unsupported claims only detract from this valid observation, and they are not in the interest of either Authors or readership.
Author Response
Reviewer #1: comments
Abstract. The Authors have found an association: it is not correct to say that the heterozygous state for an ADAMTS13 mutation ‘leads’ to a severe form of COVID-19, since in many cases it does NOT lead to this.
We agree with the reviewer that rare mutations in ADAMTS13 do not necessarily lead to a severe form of COVID-19. We modified this sentence in the previous revision as follows: “We report here that ultra-rare variants in a heterozygous state lead to a rare form of COVID-19 characterized by hyper-inflammation signs…”. This means that only rare COVID-19 cases are caused by ADAMTS13 ultra-rare variants.
Page 4. The phrase ‘One heterozygous ADAMTS13 ultra-rare variant’ gives the impression that it is always the same. I think the Authors mean: “One of several ADAMTS13 ultra-rare variants (see Table 2)….”
We modified the sentence as suggested by the reviewer.
Fig. 1. The data and their presentation are unsatisfactory for several reasons. (1) Out of 124 patients with ultra-rare variants only 11 were tested; out of hundreds of ‘controls’ only 5 were tested. (2) Of the 11 'cases' we see data on only 6; I don’t know what it means that outliers were removed: in my view there is no justification for removing any data. (3) The control group is homogeneous in being presumably wild-type for ADAMTS13; the ‘cases’ group is heterogeneous, because it contains subjects who have different mutations. (4) Of the 'cases', 4 out of 6 have the same mutation, therefore they are not representative of the multitude of mutations listed in Table 2. Thus the data are over-interpreted, not to say misleading.
(1) We thank the reviewer for this comment. Indeed, we need to give more details about the assessment analysis results, as it might seem misleading due to the small number of subjects tested. However, we highlight that we did not select any of them specifically for this paper. The functional analysis of ADAMTS13 was performed by the recruitment center on subjects for whom a fresh blood sample was available. Afterward, we checked if the subjects, for whom the ADAMTS13 assay was successfully performed, had any ultra-rare mutation in ADAMTS13. On these bases, we did the division into cases and controls. Unfortunately, for this work, only a few samples were available. In fact, the center reported that although initially 16 patients had fresh-blood draw, only 11 (6 cases and 5 controls) passed the blood draw quality check and had ADAMTS13 assay successfully performed. We rewrote the paragraph accordingly, reporting in the text only the "real" total number of samples included in Figure 1; we also better describe the method section regarding ADAMTS13 activity.
(2) As outliers we considered the values that were numerically distant from the rest of values in the “cases” dataset.
(3) We agree with the reviewer that the “cases” group is heterogeneous and not all ultra-rare variants have the same impact on protein activity. Having the activity analysis data only for a few mutations, we were not able to define the impact of every single mutation.
(4) As we explained above, the functional analysis of ADAMTS13 was not performed for this research paper, it was already done. We integrated the available data to show that the majority of carriers of ultra-rare variants had a reduced enzymatic activity in comparison to no carriers. We are aware that no robust conclusions can be drawn from this analysis because, as the reviewer said, the ‘cases’ group is not representative of the multitude of the variants identified in our cohort. We clearly stated in the end of the paragraph that we do not know how many of the heterozygotes had low ADAMTS13 levels. The variant p.P952S is present in 4 subjects because they are family members. This result is an important starting point for further analysis to better define the impact of rare ADAMTS13 heterozygous variants on protein activity.
Fig. 2. The claim of autosomal dominant inheritance of severe COVID-19 based on 2 families (with only two generations and only one child in each), in one of which it has NOT occurred, borders on absurdity. As for segregation, one needs at least two children from a heterozygote, one inheriting the mutant gene and the disease, the other the wild type gene and no disease. This section, the corresponding statement in the abstract, and the corresponding paragraph in the discussion, need to be re-written.
We thank the reviewer for this comment. Although further segregation analysis in other families will be necessary in order to confirm this mode of inheritance, we think it is not wrong to talk about “autosomal dominant condition”. The autosomal dominant inheritance described in subjects with heterozygous ADAMTS13 variants, as we described in the paper, is conditioned by SARS-CoV-2 infection, sex, and age. The 40-year-old heterozygous female, SARS-CoV-2 infected, who was oligosymptomatic is an example of how the penetrance is not complete for this disease and it is influenced by the sex and age. Moreover, we have to consider that variability in the COVID-19 phenotype could be modulated also by the presence of other genetic factors that could influence the different COVID-19 phenotype presentation. For all these reasons, we would like to maintain the term “autosomal dominant, conditioned by SARS-CoV-2 infection, sex and age” to describe this rare COVID-19 form due to ADAMTS13 rare variants.
Conclusion. At the moment caplacizumab is indicated for acquired thrombotic thrombocytopenic purpura in combination with plasma exchange and immunosuppressive therapy; crizanlizumab is indicated to reduce the frequency of vaso-occlusive crises in patients with sickle cell disease. The Authors are perfectly entitled to tentatively suggest that these drugs might have a role also in patients with COVID-19 who are heterozygous for certain ADAMTS13 mutations: but, in my view, in doing so they should be more cautious and a little more modest, especially since they have demonstrated in these patients only hyper-inflammation, not thrombophilia. The Authors are not entitled to say that these drugs “will… prevent thrombotic events”
We agree with the reviewer's comment and we rewrote the “Conclusion” paragraph accordingly.
Overall, the Authors have found an interesting association between ADAMTS13 mutations and severe COVID-19. In my view unsupported claims only detract from this valid observation, and they are not in the interest of either Authors or readership.
We rewrote the paper following the reviewer's suggestions.